# Last Mile Logistics Innovations in the Courier-Express-Parcel Sector Due to the COVID-19 Pandemic

**Łukasz Sułkowski [1], Katarzyna Kolasińska-Morawska [2], Marta Brzozowska [3], Paweł Morawski [4],\* and Tomasz Schroeder [2]**

1. Faculty of Management and Social Communication, Department of Management of Higher Education Institutions, Jagiellonian University, 30-348 Cracow, Poland; lukasz.sulkowski@uj.edu.pl
2. Department of International Management, Cracow University of Economics Management Institute, 31-510 Cracow, Poland; kolasink@uek.krakow.pl (K.K.-M.); schroedt@uek.krakow.pl (T.S.)
3. Faculty of Law and Social Sciences, Department of Management, Jan Kochanowski University of Kielce, 25-369 Kielce, Poland; marta.brzozowska@ujk.edu.pl
4. Faculty of Organization and Management, Lodz University of Technology, 90-924 Lodz, Poland
\* Correspondence: pawel.morawski.1@p.lodz.pl

**Abstract:** The development of the e-commerce market worldwide, which was already dynamic, was accelerated by the SARS-CoV-2 virus. Millions of incoming orders required analogue support from the CEP sector (courier-express-parcels sector) to provide the desired "customer experience". In the context of whether the habit of shopping in virtual reality will become permanent, it is worth considering what shape the logistics services will take in the last mile after the pandemic? Or, will customers return to shopping in the real world? A subject for these considerations was an analysis of the impact of the SARS-CoV-2 virus pandemic on the technologization of last mile logistics services, resulting in an increase in the level of "customer experience", with Poland as an example. The research methods used were participant observations and critical analysis of collected materials. The obtained results made it possible to conduct a descriptive and explanatory nomothetic study based on an Internet questionnaire. The authors formulated a diagnosis about the possibilities of using the potential of customer experience for the development of enterprises based on technologization of last mile deliveries. The recommendations can be used by scientists and managers in the CEP industry to redefine business models based on the technology of logistics customer service processes.

**Keywords:** last mile; logistics services; technologies; innovations; COVID-19

## 1. Introduction

The basis of the modern economy is the digital revolution and information management. Its special feature is the speed of providing information. So, at the end of 2019, the world was informed about the COVID-19 epidemic (this acronym was created by combining three English words and a number: CO—corona, VI—virus, D—disease, 19—year), which in a very short time was recognized as a pandemic, impacting the functioning of the global community for many months. Information on the pandemic and its effects was available from TV news links, press titles and websites. Indeed, the virus did spread during the pandemic. The total number of coronavirus infections in the world has exceeded 502,213,040 since the start of the pandemic (as of 4 January 2022) [1]. This new reality has become a considerable challenge not only for ordinary citizens.

The economies of countries, sectors, industries and individual companies were subjected to a test that not all of them passed successfully. Among many, however, there were those that resisted the restrictions and limitations. It can almost be said that they took advantage of the imposed lockdowns. When face-to-face meetings were not possible, the lives of millions of citizens moved from real to virtual space. E-commerce and logistic activities in last mile services have become those which, thanks to the use of innovative

information and communication technologies (ICT), ensured the continuity of citizens' functionality. Therefore, it is worth analyzing this redirection of transactional traffic from the real to the virtual world.

During the pandemic, sales and purchases on the Internet have become the only possible medium for marketing and participation in transaction processes. Sales dynamics grew [2]. Logistics, which has undergone a digital transformation in recent years, came to the rescue of electronic transactions. Thanks to technological progress, intelligent logistics (smart logistics) is no longer just a fashionable slogan, but a real concept implemented in modern companies fulfilling logistics processes [3]. The emerging concepts of Society 5.0 and Economy 5.0, based on innovation and technological progress, have become a catalyst for technological innovation, which in the time of the COVID-19 pandemic has turned out to be extremely useful.

In general, the requirement to adapt to the new reality meant that companies were forced to flexibly and quickly follow the changes. The solution was the increased emphasis on technology in line with the Industry 4.0 concept, which was first mentioned in 2011 at the Hanover trade fair [4]. The Industry 4.0 concept aims to integrate intelligent machines and systems, enabling the introduction of effective changes in production processes, but not only that. The goal of this strategy is to increase operational efficiency and introduce the possibility of flexible changes in the offered assortment [5]. Such activities are possible thanks to digitization and virtualization.

Despite the admiration for the concept of Industry 4.0, scientists and practitioners note [6–8] that the mere use of machines and devices, cloud computing technologies, big data and the Internet of Things to increase the effectiveness and efficiency of enterprises is insufficient. So, technology alone is not enough—people still play an important role in this process, including employees, customers and stakeholders. Therefore, it is particularly important to properly prepare members of society to meet the needs of the technologized economy [9]. Researchers and practitioners agree that people's knowledge, competencies and skills are becoming priceless, which indicates the transition to the next stage of development. Moreover, although the development and implementation of the Industry 4.0 concept is at the stage of strong growth, there is more and more talk about the Industry 5.0 concept, in which people, supported by advanced technological solutions, play a key role. However, in contrast to the focus on technology, there is a shift towards human nature according to the broader notion of Society 5.0 [10–15], which can "balance economic progress with solving social problems through a system that strongly integrates cyberspace and physical space" [16]. Additionally, the concept of Economy 5.0 has emerged and refers to collaborating on innovation, creativity, and the competitiveness of society and individuals to devise unique ways of creating value in economic structures.

In essence, e-commerce is based on the use of digital tools, and self-improvement is inscribed in the DNA of each enterprise. However, selling in digital space alone is not enough. It is necessary to finalize the purchased goods in the form of a delivery service. Logistics is essential here. Therefore, the CEP industry, which provides last mile logistics services, is consistently orbiting around the e-commerce industry [10].

The use of new technologies in the area of Industry 4.0 has created new opportunities for both CEP industry operators (current order tracking, automatic sorting, technological support for deliveries) and customers (automatic parcel sending with the use of software, shipment tracking, electronic payments, deliveries to parcel machines and many more). The innovative solutions used in the area of delivery logistics have been given a special test during the COVID-19 pandemic. Not only is timely delivery important, but also safe delivery. Customers ordering products on the Internet have the opportunity to use various delivery formulas based on innovative technological and process solutions. Therefore, it is worth analyzing the issue of customer experiences with the CEP industry services acquired during the pandemic and obtaining answers to the following questions:

- Has the black swan, i.e., the COVID-19 pandemic, influenced online shopping and to what extent?

- To what extent have the innovative solutions in the area of last mile services used during the COVID-19 pandemic been accepted by customers?
- Will the "customer experience" acquired by customers during the COVID-19 pandemic result in an increase in customer expectations regarding the technologization of delivery services offered by the CEP industry in the area of last mile logistics?
- To what extent will technology determine the future of the CEP industry?

The authors decided that Poland could represent the activity of the CEP industry in the area of last mile services. The industry of courier, express and postal services (CEP) has existed for many years. The company that started it all was the American Messenger Company, founded in Seattle, United States in 1907 by J.E. Casey, which after the transformations acquired the present name of United Parcel Service (UPS). The history of courier services in Poland is much shorter; the first Servisco company that conducted this type of activity was listed in 1982. Despite such a short history of the CEP industry in Poland, it is thanks to the use of the most modern world technological solutions that the industry is developing very dynamically.

Competition between enterprises forces the introduction of innovative solutions in the processing area and new technologies in infrastructure. Over the past 30 years [11], thanks to mutual competition, enterprises operating in the CEP industry in Poland have undergone a general metamorphosis, which is in line with global trends, as the vast majority of enterprises are corporations operating on a global scale. Thus, the changes introduced in Poland were the ones that were also introduced worldwide. Taking into account this premise, the authors defined the purpose of the article, which was to show the impact of the COVID-19 pandemic on the way of providing services offered by enterprises in the CEP industry, which represents the increase in "customer experience" in the exemplification of technological solutions in customer service at the last mile stage in the example of Poland.

Taking into account the fact that the progress of globalization in the world, the development of technology and ubiquitous computerization generate new challenges that have to be faced by service industries, enterprises must not only be reactive on an ongoing basis, adapting to market expectations, but also take proactive measures. Being online is a sine qua non condition. The area of competitive differentiation and defining the strategic differentiator is the form of interaction with the client in the service process at the stage of service implementation and accompanying service. In order to determine whether the direction of last mile logistics technologization is the right one, the authors decided to verify the thesis.

Provision of logistic services by the CEP industry is related to the necessity to increase the level of experience in service in the service-making process based on technological innovations.

In order to be able to verify the thesis formulated in this way, four research hypotheses were outlined:

**Hypothesis 1 (H1).** *The CEP industry played a significant role in the e-commerce activity in the time of the COVID-19 pandemic, determining the assessment of finalizing transactions.*

**Hypothesis 2 (H2).** *Customers using innovative solutions in the area of CEP industry services during the COVID-19 pandemic were satisfied with the last mile service formula applied.*

**Hypothesis 3 (H3).** *The experiences of customers during the COVID-19 pandemic will result in an increase in customer expectations regarding the pro-innovation of delivery services offered by the CEP industry in the area of last mile logistics.*

**Hypothesis 4 (H4).** *Innovation in the field of technologization of delivery services in the context of logistic last mile service will determine the future of the CEP industry.*

The analysis of the collected material made it possible to determine the theoretical and practical implications of the experiences of the CEP industry customers in the area

of applying innovative solutions in customer service, with exemplification in the form of creating acceptable and diversified forms of automation of delivery services in the structure of last mile logistics in the future.

## 2. Last Mile Service Supporting Transactions on E-Commerce Markets

The rapidly increasing online shopping volume generated by the COVID-19 pandemic has made delivery logistics a priority. It was not enough just to have a sales site and be present on auction portals. The issue of deliveries to the customer has become a critical issue—the so-called last mile. The fate of organizations and citizens has been dependent on technology and companies operating in the logistics services market, i.e., couriers, express parcels and parcels (CEP). Logistics service providers for online stores have redefined their action plans and have been agile in adapting to new challenges.

After more than a year, the first analyses and summaries could be made, especially in the area of innovations introduced by companies from the CEP industry in the field of customer logistics. Data from the global report "COVID-19 and e-commerce: a global review" prepared by the United Nations Conference on Trade and Development (UNCTAD) [12] indicate an increase in the share of e-commerce in global trade from 14% in 2019 to nearly 17% in 2020. Increasing tendencies are also indicated by the report "E-commerce in Poland—strategies for the development of companies" by Mazars, Noerr and SpotData, according to which the Polish e-commerce market in 2020 was worth PLN 70 billion [13]. With the annual increase of several percent so far, the time of the pandemic has turned out to be record-breaking with a peak year-on-year (2019 to 2020) increase of 43%. Moreover, although specialists are of the opinion that after the pandemic the increasing tendency will slow down [14], the habits of online shopping will certainly remain both in the world and in Poland. This heralds the further development of the courier services market, which as a representation of last mile logistics, is qualified as a source of potential advantages in the fight for an increasing share of the e-commerce market. It is this final stage of distribution and the point of contact between the customer and the seller [15] that is very often decisive and will determine in the future the general impressions of buyers who, by posting positive or negative opinions about sellers on the Internet, may decide whether they survive or not.

The customer is the basis for the functioning of the service business. Therefore, enterprises must constantly and very carefully take care of building the value of their product or service in order to be able to strengthen their market position and increase their competitiveness in relation to other companies in the service sector. It is no longer enough to meet the customers' conscious expectations, but to exceed them [17].

In addition to knowing the individual components of customer service at the pre-transactional, transactional and post-transactional stages, it is also necessary to be aware that clients are becoming more and more demanding, and numerous sources of competition make customer service a distinguishing feature of enterprises, contributing to their success or failure [18]. Professionalization of logistic customer service requires standardization, which will ensure both the distinction and highlights of the company's offered services, i.e., a kind of uniqueness. Therefore, standardization includes systematized rules and guidelines that give the possibility of determining specific behaviors that should be adopted by employees in contact with clients. The standards show the values followed by the company and define the direction the company wants to follow [19]. Customer service is primarily built on activities aimed at developing service standards, order fulfillment and after-sales service [20]. It is a strategic approach that is part of the company's overall strategy. Thanks to logistic customer service, it is possible to create a differentiator based on the core of logistic skills supported by service components that allows the company to distinguish itself from the competition and achieve market success.

Standardization is to guarantee the repeatability of the service process strictly in accordance with an established procedural pattern so that the customer is guaranteed that he will be able to count on the same experience each time. In order for service standards to be maintained, they must first be set. For this purpose, enterprises carry

out customer preference surveys, the results of which are guidelines for the desired level of service specified by customers. They include both interpersonal (soft) and executive-logistic (hard) standards. Based on the data obtained, a set of rules and a standard for the implementation of the customer service process are constructed. Then, their testing and target implementation are carried out [21].

The consequence of the development of e-society is the emergence of a new concept describing a completely new type of consumer that business is dealing with, namely the e-consumer [22]. The distinguishing features of the e-consumer are that he is, above all, well-informed, mobile and educated. Independently and consciously, he decides what, how and where to buy. Moreover, the e-consumer is a demanding client with a low tolerance for errors made by service providers. A characteristic feature, however, is that the e-consumer satisfies his needs by making online purchases [23].

Adjusting the market to the requirements of consumers has increased the importance of logistic customer service [24], which oscillates around digitization, as never before. The growing demand from customers coming from e-commerce has made courier services not only a means of convenient shopping, but also a method of safe shopping [25]. Sanitary guidelines have moved the processes of delivering courier services towards digitization. Enterprises have implemented additional security procedures and introduced many innovative solutions that ensure digital possibilities for the delivery of services in line with reported demand. The implementation of non-cash payments, electronic signatures, confirmations with the use of one-time PIN codes, parcel machines and refrigerators are just some of the solutions. The resulting innovations, if they proved successful at least once, should be permanently implemented to drive socio-economic growth, which will be permanent, repeatable and significant.

## 3. Unforeseen Events—SARS-CoV-2 As Destructor and Inductor

The business world is often faced with crisis situations. These can be short-term, sudden situations with little impact on the functioning of the enterprise. However, when an event occurs that destabilizes processes for a long time, then new concepts are necessary that will support the daily functioning of enterprises. It often happens that crisis events result from the macro-environment in which international factors play an important role, especially for companies from the logistics industry. Each of the elements of the macro-environment can be treated as a source of opportunities or threats, in line with the theory of economics (SWOT/TOWS, PEST analysis).

Of course, crises, from the point of view of business and social life, can mean many situations. The terrorist attacks in the USA in 2001, which influenced the aviation industry and compelled it to change procedures and increase passenger safety, could be considered in such context [26,27]. Another type of crisis unfolded with events in the financial markets in 2007–2009, which were described as a global financial crisis caused by the US mortgage bubble and the collapse of Lehman Brothers bank [28,29]. Crises can also be viewed in terms of natural disasters, such as the 2011 earthquake and resultant tsunami in Japan [30]. All of these events have an impact not only on the place where they started, but also, like the tsunami wave, cause crises and changes at subsequent points on the world map in the social, economic and even political dimensions.

An important issue is the ability to deal with such situations. Searching for the best solution, especially in situations related to business or the national economy, requires the use of appropriate methods and crisis management tools. Crisis management activities can be divided into two categories: avoiding and combating crisis situations by taking compensatory measures. Unfortunately, defensive actions are often overdue responses. Therefore, opportunities should be sought in anti-crisis measures that will make it possible to avoid a crisis or to plan such measures that will allow us to survive the crisis with the least possible loss. The anti-crisis measures can be divided into two phases. The first is preparation for the crisis, i.e., planning appropriate actions, setting up response systems, and simulating and training staff so that they know what procedures to follow. The second

phase involves actions taken during a crisis that could not be avoided. At this stage, first of all, warning systems using data analysis should be used. You should also control the operation of the entire organization and all internal systems. With the use of such tools, it is possible to manage risk and react in an emergency, counteracting undesirable events.

However, what to do when an incident occurs that appears to be unpredictable? In the literature, crisis events that are treated as unpredictable are referred to as the so-called black swans. In relation to market issues, "black swans" are sudden shifts in volatility that affect stock markets, industries, and even entire economies [31]. The SARS-CoV-2 pandemic is considered by many as the "black swan". Still, more than two years after the first restrictions and limitations, many economies, even the most developed ones, feel the effects of the lockdowns. Crises related to the availability of raw materials and components are noticeable. The most visible and noticeable crisis for many manufacturers is the crisis related to the availability of semiconductors used in the automotive, electronics and computer industries [32–34]. Fortunately, more and more countries have decided to limit the restrictions; however, such steps are conditioned by anti-crisis measures in a pandemic situation (i.e., high rates of vaccination in the population of a given country).

The questions remain, however. Was the SARS-CoV-2 pandemic really a "black swan"? Was it an unpredictable situation? Are other economic crises, natural disasters or those caused by bad political decisions unpredictable? Many scientists assume that crises are a natural part of socio-economic life. It is important to be able to draw conclusions from them and learn how to use crisis situations to create an opportunity for the development of innovative concepts [35]. Nicknamed "Dr. Doom" [36,37], in 2005 Prof. Ariel Roubini predicted a financial crisis in 2007 [38]. Another crisis was also predicted by Prof. Roubini, who in the pages of Project Syndicate together with Italian economist Brunella Rosa, pointed to the coming recession in 2020 [39]. Of course, these predictions did not concern the coronavirus pandemic, but the economic situation in the world. The pandemic has exacerbated the effects of the anticipated crisis. On the other hand, predictions about a pandemic also appeared, for example, after the simulation tests of the Institute of Disease Modeling and the research of Vaclav Smil. These predictions were aired by Bill Gates, among others, in many of his speeches [40]. An important issue for all of the aforementioned predictions was the conclusion that the governments of countries, even the most developed ones, are not prepared for this type of recession, which suggests that the crisis management mechanisms are not fully used. Even more so, more than two years after the outbreak of the pandemic, it can be said that most countries opted for crisis management solutions. It was not until the vaccines became available that new waves of new mutations of the virus began to be counteracted. So, the governments started to use prevention.

A pandemic or financial crisis is an obvious threat to the economy and the social and political situation. Destabilization is inevitable. However, a thorough analysis of the mechanisms of crisis dynamics and implementation of appropriate measures to combat the crisis as well as preventive measures may prove to be a situation initiating positive effects.

The same can be said about the black swan, SARS-CoV-2. It had a destructive impact, primarily socially, economically and psychologically. However, there are industries that have shown the world that it is possible to come out of such situations unscathed. First of all, we should appreciate the scientists who, for the first time in history, managed to develop and patent vaccines against severe SARS-CoV-2 infection in such a short time. Another example may be the health services around the world, which have shown that, indeed in emergency situations, they can engage with all of their potential to protect the health and life of citizens.

There are also examples of enterprises that, perhaps to a lesser extent, protected human health and life, but also thanks to them, it was possible in some sense to function normally in some industries. Such an example is the broadly understood logistics industry, which has proved that, thanks to it, it is possible to maintain the continuity of the supply of food, medicines, security measures, and other products that are delivered to homes. Internet shopping was possible thanks to couriers—for shorter and longer distances. They delivered

medicines and meals from local stores or nearby restaurants, but also, the turnover of online stores increased. It can, therefore, be concluded that even in the most unpredictable situations, positives can be found.

## 4. Acceleration of Innovation in Logistic Customer Service

Crisis situations trigger further important steps that enable the implementation of new, surprising and innovative solutions. In addition, the pandemic created many new opportunities resulting from the applicable legal regulations, restrictions and trade limitations. The possibilities of building new delivery solutions for consumers that utilize the e-commerce industry are such an example. Handling last mile processes for online orders has proved to be one of the biggest challenges, especially for the trading world.

Logistics in the time of a pandemic had to take care of two aspects of safety. On the one hand, it was necessary to maintain the continuity of supply chains, especially for food, drugs, and security measures [41]. On the other hand, safe home deliveries have become important—the automation of certain processes and activities in order to avoid contact, keep social distance, and at the same time deliver parcels in accordance with accepted procedures of quality of service [42,43].

Internet shopping has become an important aspect, which made it possible to maintain existing habits and limited social contacts that could support the spread of the virus. Hence, a significant increase in sales was observed on the e-commerce market, which was still developing steadily before the pandemic. Data from the global report "COVID-19 and e-commerce: a global review" prepared by the United Nations Conference on Trade and Development (UNCTAD) confirm this state of affairs. Thus, the commercial processes required support from the logistics services. All goods purchased in the virtual world had to be delivered to their buyers. Hence, the CEP industry has gained vital importance. The growing demand from customers coming from e-commerce has made courier services not only a means of convenient shopping, but also a method of safe shopping [25]. The pandemic not only increased the importance of the digitization of the transaction process itself, but also digitized the process of logistic customer service [44,45]. Restrictions aimed at maintaining social distance, limiting contact, but also maintaining digital security, meant that logistics operators, including companies from the CEP industry, had to apply additional, coordinated measures that would protect not only their interests, but above all employees and customers.

The necessity to use distance and some disadvantages of home deliveries, also noticeable before the pandemic, meant that new opportunities appeared that developed into a new market—OOH (out of home). A common problem that appeared in many studies was the recipient's availability at the time when the parcels were delivered by couriers to the indicated address, usually home. Unfortunately, the working hours of the couriers are the same as those of all others, so, most often the recipient was not at home. A work address was often indicated, but not all recipients are able to do this. An additional option was deliveries during non-standard hours; however, from the point of view of courier companies, they were associated with additional costs, which were most often incurred by the recipient. Hence, they had the idea of allowing customers to pick up their parcels at selected times from a designated location. OOH deliveries are divided into two groups. The first option is delivery to automated parcel machines (APM), and the second is pick-up delivery (PUDO). The Out-of-home delivery in Europe 2021 report [46] shows that 40% more new PUDO points have been created in the EU since mid-2019, mainly due to the growing importance of the e-commerce market and the need to organize parcel delivery. At the same time, Poland has become a place where PUDO points increased by 70% in 2020 compared to 2019. It was probably also the effect of the Sunday trade restriction and the exceptions that allowed postal offices to operate. Hence, many retail chains decided to create PUDO points in their stores, enabling them to conduct commercial activities until February 2022.



The innovation in APMs was also conditioned by the use of additional applications for smartphones, enabling the pickup of parcels without touching the panel on the machine, which, of course, was a very good option in terms of staying safe during the pandemic. In addition, the applications allow customers to track shipments and change the pickup point, but also provide appropriate pickup codes, preventing packages from being picked up by unauthorized persons. The development of the use of parcel machines was also observed before the pandemic in various countries, as shown by research [47–50]. Of course, solving the last mile problems with parcel machines also highlights other aspects of their operation. There are also considerations about the superiority of parcel machines, or collection points in general, over home deliveries for economic and environmental reasons [51]. One can consider here the energy consumption of parcel machines and the need to use renewable energy sources so that we can talk about greater care for the environment. However, at the same time, it should be remembered that the courier does not generate additional miles when delivering parcels to one point. It is also common for recipients to choose machines closest to their home, or located on the way home, to make it easier to pick up parcels, which also helps to reduce gas emissions. Similar conclusions may apply to other collection points, located in shops, gas stations, or kiosks.

The innovation with this type of solution is certainly related to the possibility of using various applications, which, as already mentioned, facilitate the management of shipment pickup. However, last mile logistics operators are constantly looking for even more innovative solutions that will automate the delivery process. The literature review shows that most often these are the solutions that will also guarantee the use of the concept of sustainable development [52,53].

A very innovative solution is the use of autonomous vehicles/robots in handling last mile deliveries. Such robots can travel on roads [54], but they can also be drones [55,56]. The use of robots has been observed for a long time, mainly due to the interest in Industry 4.0, which is characterized by the use of robotization in various processes [57]. Robotization in the case of last mile processes focuses primarily on the use of autonomous vehicles that will be able to deliver parcels to the indicated place on their own [58]. The use of autonomous vehicles in last mile deliveries differs depending on the ingenuity of CEP operators, while the awareness among recipients of this type of service is also important. As shown by the research of Kapser et al. [59], factors such as trust in technology, price sensitivity, innovation, expected performance, hedonic motivation, social impact and perceived risk determine the perception of new ways of delivering shipments.

Drones are an even more innovative solution in handling last mile deliveries. However, it can be observed that they have more limitations in use, especially for safety reasons [60]. The limitations are the laws, restrictions on the movement of aircraft (which drones are considered to be) in certain locations, as well as cost effectiveness or simply battery durability [61]. Of course, such restrictions are a challenge, especially in urbanized spaces. On the other hand, many logistics operators are considering the use of drones in areas that are difficult to access, for example in life-threatening situations or the need to accelerate deliveries for health reasons [62]. It is definitely still a song of the future, but these are probably new, innovative areas that should be worked on.

In addition, works that would connect drones, autonomous vehicles and robots in the last mile logistics processes are being monitored all the time [63]. Certainly, all of these ideas bring logistics closer to the application of the principles of sustainable development. They allow for a broader look at the issue of last mile deliveries. Although the pandemic has sparked new ideas, and it has accelerated innovation in the implementation of IT systems in the last mile service, it is still necessary to work on new solutions that will respond to further socio-demographic problems and, for example, will be adapted to the needs of people representing the so-called silver economy [64] or bottom of pyramid (BOP) [65].

## 5. Materials and Methods

Primary (fragmentary) explanatory research of a descriptive and illustrative nature, focused on the implementation of one research goal, was carried out at the turn of 2021 and 2022 in Poland. The goal was to determine the impact of innovation in last-mile logistics services on the choices of customers in the e-commerce industry and predict the development of the CEP industry, taking into account the challenges of sustainable development and Society 5.0 in Poland. The researchers wanted to find out what criteria are used by individual customers buying online when choosing a logistics company offering deliveries in the area of last mile logistics. Due to the adopted type of primary source from which the data were obtained, and consequently the type of primary data obtained, a diagnostic survey was used as a research method. The indirect survey was carried out using a computer assisted web interview (CAWI) "user-centric" internet survey using the eBadania (https://ebadania.pl, accessed on 24 March 2022) platform. The measurement tool used was a questionnaire consisting of 43 closed questions, including 14 based on the scaling of attitudes according to Rensis Likert. Due to the scope of the measurement, the conducted study was fragmentary and deterministic, representative for the studied population.

The empirical material was obtained from people aged 16 and over who made purchases on the Internet at least once and used a logistics delivery service in the 3 months preceding the survey. Not all people buying online at the turn of December and January 2021/2022 were tested, but only a selected sample.

$$n_{\min} = \frac{NP(\alpha 2 \cdot f(1-f))}{NP \cdot e2 + \alpha 2 \cdot f(1-f)} \tag{1}$$

where:

$n_{min}$—is the minimum sample size
$NP$—the size of the study population
$\alpha$—confidence level for the results
$f$—fraction size
$e$—assumed maximum error

Assumptions were made about the representativeness of the sample, defining the population of Poland aged 16 and over as the studied population, which as of 31 December 2020 totaled 31,811,795 people. Taking into account the criterion of frequency of Internet use, the sample included those inhabitants of Poland who use the Internet at least once a week to search for information about goods and services there. Using the proportions of the previously conducted studies, the fraction factor was assumed as 0.814 for people aged 16 and over who regularly use the Internet (at least once a week) and search for information about goods and services there, and 0.186 for the others with a random error of 5% and a confidence level of 0.95. After making the calculations, the minimum sample size was $n_{min}$ = 233 units. The recruitment of study participants was carried out on the basis of the non-random selection of units typical for the snowball test, which means that the request to participate in the study was forwarded via social networks to all persons meeting the conditions of the study participant: age and act of making purchases at least once and used a logistics delivery service in the 3 months before the test.

After the respondents filled in the questionnaires, the obtained raw data were encoded in SPSS version 20, Poland, Cracow (Statistical Package for Social Sciences) program using a code key. After data verification and validation, a three-stage data analysis was performed. In the first stage, a comparative analysis of the population and the studied sample in the area of comparative characteristics was carried out using the non-parametric significance test, which is the compliance test based on the $\chi^2$ statistics. Then, the distributions of the obtained results were presented. In the third stage, where it was possible and logical and statistically significant, the hypotheses on the relationships between the selected variables were verified.

Thus, at the beginning, the null hypothesis (H0) was adopted about the compliance of the distributions of selected variables (sex, age, place of residence) from the sample with the

distributions characterizing the population of Poland aged 16 and over, and the alternative hypothesis (H1) about the existence of such inconsistency.

$$\chi^2 = \sum_{i=1}^{r} \frac{(n_i - np_i)^2}{np_i} \tag{2}$$

where:

$p_i$—denotes the probability that the feature $X$ will take a value belonging to the class range

$np_i$—denotes the number of units that should be in the i-th interval, assuming that the feature has a distribution consistent with the hypothesis.

The statistics, on the other hand, have a distribution $\chi^2 \alpha o k = (r − 1)$, where:

$k$—denotes the number of degrees of freedom

$r$—is the number of class ranges

$\chi^2$—is the empirical value of the statistics obtained from the study.

The form of the critical set: $P (\chi^2 < \chi^2 \alpha) = \alpha$, where $\chi^2 \alpha$ is the critical value from the distribution tables, $\chi^2$ *dla* $k = r − 1$ degree of freedom, and $p = \alpha$.

Ultimately, more than the necessary minimum number of respondents participated in the study, $n_{min}$ = 233 units, i.e., 658 respondents. Women (53.2% of the respondents) dominated in the sample of participants in the study compared to men (46.8% of the respondents). Taking into account the age of the respondents, the largest group (36.0%) comprised people aged 55 and over. The remaining respondents were aged 35–44 (19.3%), 25–34 (16.3%), 45–54 (16.1%) and 16–24 (12.3%). People under the age of 16 were not included in the process of sampling and conducting the survey due to legal restrictions on persons who can make electronic payments. In turn, taking into account the place of residence, four out of ten respondents (40.1%) lived in the countryside, and nearly six out of ten in cities (59.9%).

As a result of the analysis, it was possible to confirm the null hypothesis (H0) of the existence of consistent distributions, i.e., the distribution of variables characterizing the studied sample was consistent with the study population of Poland in terms of sex, age and place of residence (Table 1).

**Table 1.** Sample representativeness—Statistical concordance test $\chi^2$.

| Parameters | Number | Number | Value $\chi^2$ Real | Value $\chi^2 \alpha$ Theoretical | Test Realization $\chi^2 < \chi^2 \alpha$ |
|---|---|---|---|---|---|
| **Sex** | | | | | |
| Female | 350 | 16,722,685 | | | |
| Male | 308 | 15,089,110 | 0.103 | 3.841 | concordance |
| **Age** | | | | | |
| 16–24 | 81 | 3,395,513 | | | |
| 25–34 | 107 | 5,222,883 | | | |
| 35–44 | 127 | 6,300,861 | 2.204 | 12.592 | concordance |
| 45–54 | 106 | 4,975,279 | | | |
| 55 and above | 237 | 11,917,259 | | | |
| **Place of residence** | | | | | |
| Countryside | 264 | 15,359,918 | | | |
| City up to 20 k | 70 | 4,983,795 | | | |
| City from 20 k to 49 k | 72 | 4,237,100 | | | |
| City from 50 to 99 k | 60 | 3,128,500 | 4.197 | 11.07 | concordance |
| City from 100 to 199 k | 63 | 3,324,500 | | | |
| City above 199 k | 129 | 7,231,300 | | | |

$\alpha$—confidence level.

In the selection of individual participants for the study, non-random sampling was used; therefore, it was difficult to fully assess the representativeness of the samples in a statistical sense due to the lack of randomness. However, due to the size of the received sample, it was possible to infer.

Therefore, after analyzing the structure of distributions of answers from individual basic questions, a cross-analysis was carried out, allowing us to determine the possibility of the existence of relations between the selected variables. The basis at this stage was the contingency table each time, allowing for the combination of two features at the same time. The table consisted of r rows and s columns each time. Each row and column corresponded to particular variants of the feature $X$ and $Y$. The content of the contingency table consisted of the nij numbers of sample elements that have the *i*-th variant of the feature $X$ ($i = 1, 2, \ldots, R$) and the *j*-th variant features of $Y$ ($j = 1, 2, \ldots, s$). Each time the contingency table was the basis for the verification of the null hypothesis (H0) of the existence of the potential stochastic independence of the random variables $X$ and $Y$ and the alternative hypothesis (H1), adopted when the null hypothesis (H0) was rejected, according to the formula:

$$
\begin{aligned}
H_0 : \ & P\{X = x_i \ \wedge \ Y = y_j\} = P\{X = x_i\} \quad P\{Y = y_j\} \\
H_1 : \ & P\{X = x_i \ \wedge \ Y = y_j\} \neq P\{X = x_i\} \quad P\{Y = y_j\}
\end{aligned}
\tag{3}
$$

The basis for the verification of the H0 hypothesis about the stochastic independence of variables was the value of the statistics obtained from the formula:

$$
\chi^2 = \sum_i^r \sum_j^s \frac{\left(n_{ij} - \widetilde{n}_{ij}\right)^2}{\widetilde{n}_{ij}} : \chi^2_{(r-1)\cdot(s-1)}
\tag{4}
$$

where:

$n_{ij}$—conditional empirical numbers resulting from the contingency table,

$\widetilde{n}_{ij}$—theoretical conditional counts that could appear in the table if the features were independent.

Hypothetical numbers are determined according to the formula:

$$
\widetilde{N}_{ij} = \frac{n_{i\cdot}\cdot n_{\cdot j}}{N}
\tag{5}
$$

The H0 rejection area is always right-handed. Its size depends on the adopted significance level $\alpha$. It is greater the greater that $\alpha$ is. Generally, $\alpha \leq 0.05$ is assumed. Critical values of the $\chi^2$ distribution with $(r - 1) \times (s - 1)$ degrees of freedom. If only $\chi^2$emp $> \chi^2\alpha$, then H0 is rejected in favor of the H1 hypothesis, which means that the pair of features is mutually dependent on each other.

Using the above methodology, the process of testing the statistical significance of the relationship between the selected variables was carried out. Technically, the SPSS computer program was used to carry out the calculations. Each time, the verification of statistical significance for the indicated variables consisted of checking whether the value of the asymptotic significance parameter for the $\chi^2$ statistical value of a given pair of analyzed variables was lower than 0.05. If so, the observed relationship between the variables could be considered statistically significant, and the results and conclusions drawn on the basis of the analysis of data obtained from the sample were considered representative and could be transferred to inferences about the entire population.

## 6. Results

The conducted research allowed for the analysis of the material in the areas of availability and types of supply logistics services, evaluation of the functionality of innovative forms of supply logistics services and the indication for future solutions and infrastructural components in servicing last mile logistics. On the basis of the collected material, a prediction was made of the development of innovative customer service formulas in the future for the technologization of the last mile logistics service, resulting in an increase in the level of "customer experience" using the example of Poland.

*6.1. Choosing the Internet for a Shopping Destination*

Participants in the study stated that when deciding to buy goods in digital reality, they made purchases in this way occasionally, most often by using the services of an exclusive online store (61.7%) and an e-shop of a stationary network (58%, 4%). Less frequently, the respondents indicated that e-auction services (40.7%) and sales platforms (40.4%) were occasionally chosen as shopping places. The least frequently chosen occasional shopping places were social networks (26.7%).

Taking into account the frequency criterion, every fourth respondent who purchased at exclusive online stores (25.2%) and every fifth buyer from e-shops of brick-and-mortar networks (21.3%) and e-auction websites (17.9%) decided to buy once or twice a month. The smallest number of respondents indicated that they decided to make purchases once or twice a month using the internet platform (12.2%) and on social networks (2.4%). The respondents who chose exclusive online stores (10.1%), sales platforms (7.3%) and e-shops of stationary networks (6.1%) decided to buy more often in the digital space, i.e., one or two times a week. The least frequent purchases once or twice a week were through social networking sites (3.9%) and e-auction sites (2.4%).

Respondents buying at exclusive online stores stated that during the SARS-CoV-2 pandemic, they mostly chose these units for shopping twice as often (51.4%). A similar choice was made by every fifth respondent buying at e-shops of the stationary network (20.7%) and using the services of e-auction services (20.4%). Only one in ten respondents buying on a sales platform (15.8%) or on a social networking site (11.6%) stated that they made transactions twice as often. The majority of respondents who used stationary e-shops (53.2%), and every third respondent using a sales platform (36.8%), an exclusive online store (38.6%) or an e-auction service (36.2%) declared that they made purchases through selected digital trade formats on the same level as before the pandemic.

The respondents, when asked about the reasons for choosing digital formats for shopping sites, stated that the main criteria were definitely a large selection (96.0%), access to products that are not available locally (95.7%), convenience, meaning access to the 7/24/365 services and possibility of returns (93.9%), time savings in reaching the store (90.3%) and safety and health protection (65.0%).

Large selection as a contribution to online shopping was indicated more often by women (100.0%) than men (91.6%), people younger than 34 (over 70%) and residents of cities with over 50,000 inhabitants (over 90%). Access to products that were not locally available contributed to online shopping more often for women (100.0%) than for men (91.0%), people younger than 34 (over 75%) and residents of cities with less than 50,000 inhabitants and villages (over 90%). The convenience of online shopping was indicated more often by men (94.8%) than women (93.1%), people younger than 34 years of age (over 60%) and residents of cities with more than 50,000 inhabitants (over 90%). Saving time was more important for women (90.9%) than for men (89.6%), people younger than 34 years of age (over 80%) and residents of cities with over 50,000 inhabitants (over 90%). On the other hand, in terms of safety related to health protection, online shopping was more often chosen by women (77.1%) than men (51.3%), older people over 45 (over 60%) and residents of cities with over 50,000 inhabitants (over 80%). The existence of the indicated relationships was confirmed by the $\chi^2$ independence test, with the strength of the relationship determined by V-Cramer (Table 2).

Such results can be summarized by the statement that, according to the declarations of the respondents, digital sales formats, due to their value, were eagerly chosen for shopping places, and the pandemic contributed to the increase in the dynamics of this phenomenon. Becoming familiar with the services and attributes of e-commerce allows one to assume that those who have had positive experiences using digital transactions at least once will renew them, which will contribute to an increased turnover of goods in the virtual space.

**Table 2.** Reasons for choosing the Internet as a shopping destination by gender, age and place of residence—$\chi^2$ independence test with relationship strength determined by V-Cramer.

| Variants | Sex | | | Age | | | Place of Residence | | |
|---|---|---|---|---|---|---|---|---|---|
| | $\chi^{2\,a}$ | $p^{\,b}$ | $V^{\,c}$ | $\chi^2$ | $p$ | $V$ | $\chi^2$ | $p$ | $V$ |
| Convenience (7/24/365 access and the possibility of returns) | 104.952 | 0.001 | 0.330 | 62,065 | 0.001 | 0.179 | 325,112 | 0.001 | 0.410 |
| Large selection | 59.495 | 0.001 | 0.248 | 100,939 | 0.001 | 0.229 | 201,128 | 0.001 | 0.323 |
| Access to products that are not available locally | 48.846 | 0.001 | 0.225 | 122,619 | 0.001 | 0.206 | 241,577 | 0.001 | 0.289 |
| Saving time to reach the store | 181.368 | 0.001 | 0.433 | 79,308 | 0.001 | 0.165 | 253,947 | 0.001 | 0.293 |
| Safety—health protection | 112.318 | 0.001 | 0.341 | 96,293 | 0.001 | 0.182 | 238,089 | 0.001 | 0.282 |

[a] $\chi^2$—test value; [b] $p$—asymptotic significance; [c] Relationship strength calculated using V-Cramer.

### 6.2. Availability of Types of Delivery Logistics Services and Reasons for Using Them

Concluding transactions in the digital space in accordance with the rules of the strategy and techniques of customer service should be finalized with the transfer of products to customers who purchased products via the Internet. Here, it becomes necessary to choose a logistics delivery service. Courier services and out-of-home/out-of-work (OOH) category deliveries are now available in the last mile vendor selection area.

Among the available logistics services, the vast majority of respondents (97.0%) had contact with various forms of parcel collection from a courier, while the vast majority used it occasionally (40.7%) or twice a month (22.5%). On the other hand, significantly fewer respondents used the service of sending a parcel via courier. This form was used by slightly more than half of the respondents (60.2%), while the vast majority used it only occasionally (33.7%). See Figure 1.

The respondents had slightly less experience with the field of logistics services in the scope of using the possibilities offered by modern out-of-home/out-of-work delivery formats, i.e., a parcel locker as the place for parcel collection. Only eight out of ten respondents (79.3%) used this format, while the vast majority used it occasionally (31.9%) or twice a month (18.5%). On the other hand, the service of sending a parcel with the use of a parcel locker was used by slightly fewer participants in the study, i.e., seven out of ten (68.7%). In the case of collecting a parcel from a parcel locker, the respondents most often used the option of sending a parcel to a parcel locker occasionally (29.2%) or once a month (18.8%). Another service offered by parcel machines, i.e., the possibility of transferring parcels between parcel machines, was used by slightly more than half of the respondents (55.0%), choosing this service most often occasionally (30.7%).

The next type of delivery logistics services are deliveries to an indicated pickup/drop-off point (kiosk/shop/gas station/CEP company branch). This type of service was used by seven out of ten respondents (71.7%) most often occasionally (34.3%), once a month (19.1%) or twice a month (15.8%).

In addition to the forms of delivery logistics that have been present on the market for some time and with which the respondents were familiar, innovative format such as refrigerated parcel lockers, deliveries and dispatches by drones, and deliveries using autonomous vehicles are becoming more and more frequent. The obtained results confirm the general trend, where these solutions are currently in the market exploration phase and the market itself is adapting to them and offering new solutions. A small percentage of respondents had contact with the collection of parcels using refrigerated parcel lockers (8.2%), deliveries and dispatches by drone (4.9%) and deliveries by autonomous vehicles (8.2%).

In this trend, when analyzing the knowledge of courier companies offering last mile logistics services, the most recognizable company that respondents have heard about and whose services are known is Inpost (83.9%). The next most familiar companies indicated by the respondents were DPD (47.8%), DHL (73.6%) and Pocztex (59.9%). Geis (63.2%), Pekaes (56.5%) and TNT (43.8%) were definitely among the least recognizable companies that respondents had not heard about and were not familiar with their services. Among

companies offering courier services, the respondents also indicated those they had heard of but had no knowledge of their services. The companies mentioned were FedEx (49.2%), GLS (45.9%) and UPS (36.5%).

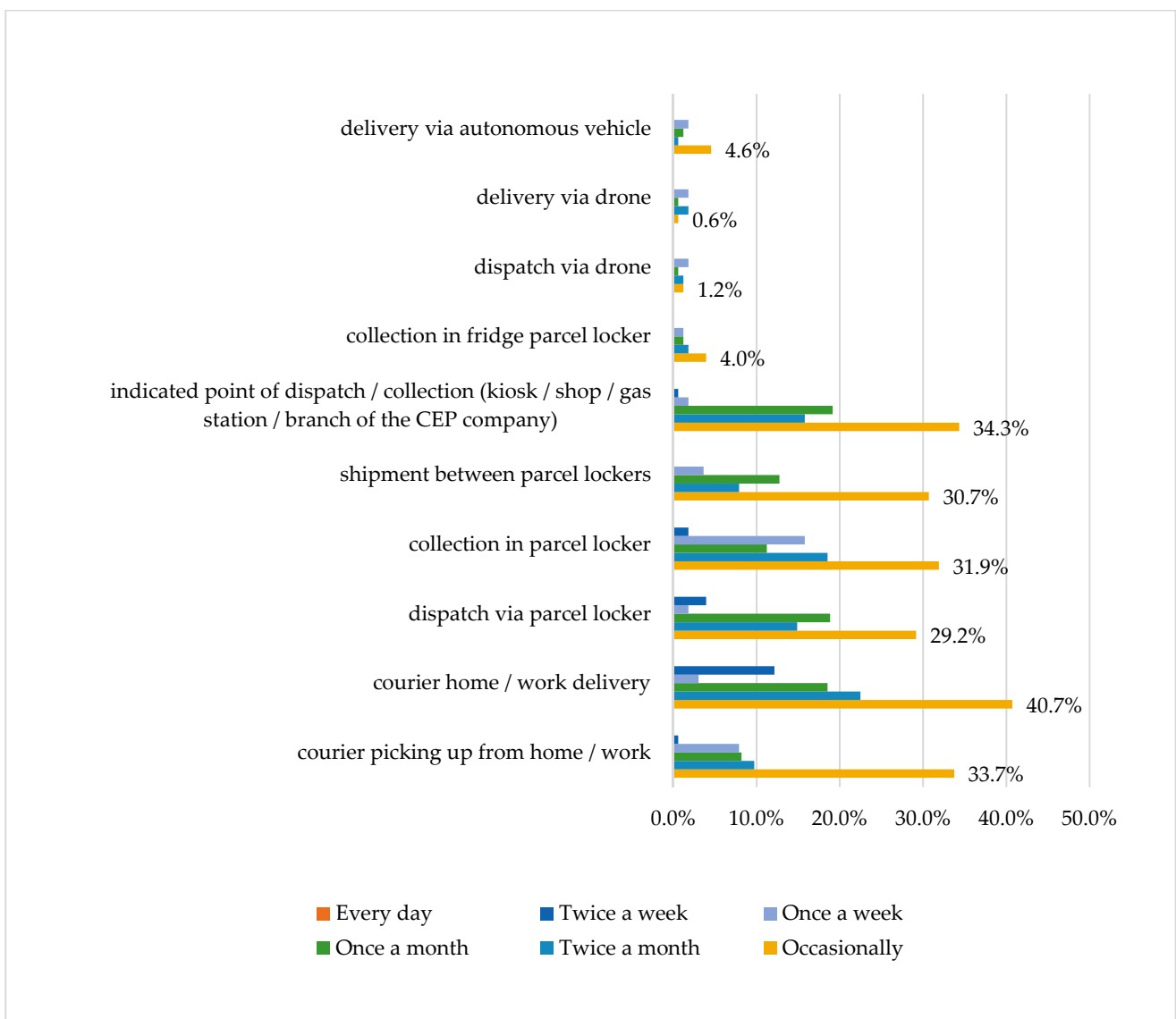

**Figure 1.** Use of last mile delivery logistics service formats.

On the other hand, taking into account the knowledge of OOH (Out of Home) logistics operators, such as the post office, retail outlets and parcel machines, the most recognizable company that respondents had heard about and whose services were known was Inpost (85.7%), followed by Allegro (69.0%), DPD (67.8%) and Poczta Polska (61.4%). BliskaPaczka.pl (55.0%), Aliexpress (22.5%) and Amazon (21.9%) were definitely the least recognizable companies that the respondents had not heard about and did not know about their services. Among the companies offering OOH services, the respondents also indicated those they had heard of but did not know about their services. The indicated enterprises were Kioski Ruchu (55.3%), Orlen (48.6%) and Żabka (44.4%).

On the other hand, respondents stated that among logistics service operators who offer deliveries using PUDO (pick up drop off) points, such as parcel machines, the most recognizable company they had heard of and whose services they were familiar with was Inpost (88.4%). The next ones indicated by the respondents were Allegro (67.5%), DPD (62.6%) and Poczta Polska (48.0%). By far the least recognizable companies operating in

Poland that the respondents had not heard of and whose services they were unfamiliar with were BliskaPaczka.pl (54.1%), Amazon (36.8%) and Aliexpress (22.5%). Among the companies offering PUDO services (parcel machines), about whom the respondents indicated they had heard of but were unfamiliar with their services, were the enterprises Żabka (67.2%), Orlen (54.1%) and Poczta Polska (50.2%). See Figure 2.

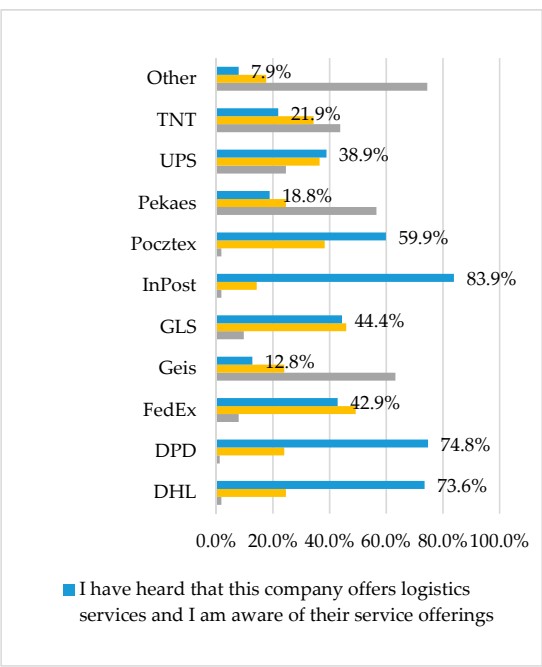

(**a**) Traditional CEP courier deliveries.

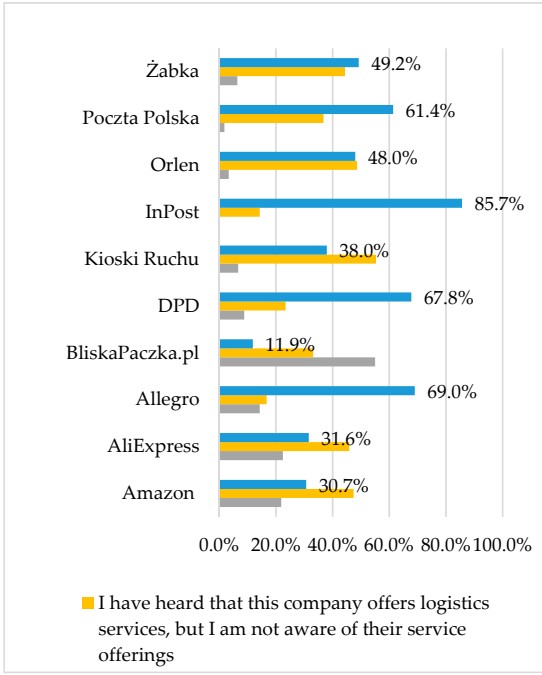

(**b**) Modern CEP OOH deliveries.

**Figure 2.** *Cont.*

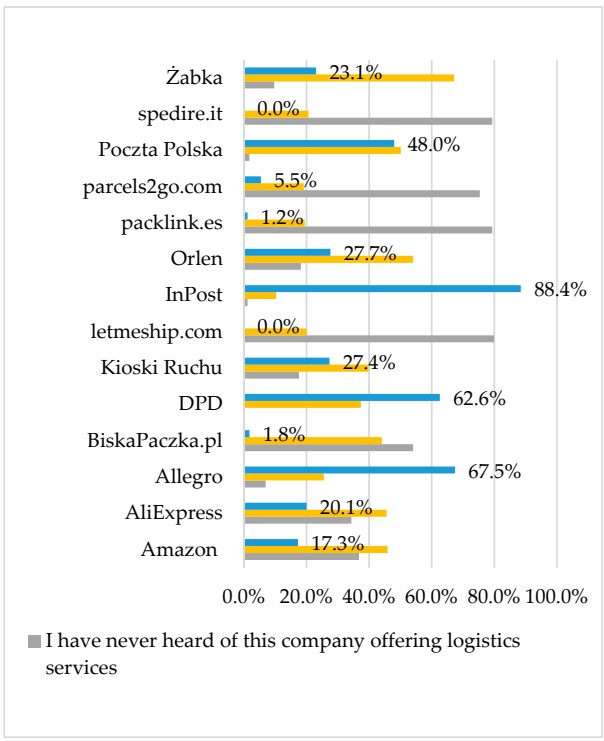

(**c**) Modern CEP PUDO deliveries via parcel machines.

**Figure 2.** Traceability and use of last mile delivery services offered by CEP companies, broken down into courier companies, out-of-home/work logistics and parcel machines.

To sum up, the undisputed, recognizable leader, recognizable by name and service offerings, is InPost, both as a traditional courier company (83.9%) and a modern OOH service provider (85.7%) including PUDO (88.4%)—automatic parcel machines. It is a company that effectively tries to combine the traditional and modern approach in the implementation of the last mile delivery service, which has been rewarded by the market with recognition. Other enterprises, apart from DPD, rather have an established image or position as providers of traditional courier or modern services, which indicates a stable mental distinction among consumers of CEP industry services.

*6.3. Analysis of the Evaluation of the Functionality of Innovative Forms of Delivery Logistics Services—Parcel Machines in the Time of the COVID-19 Pandemic*

Many of the respondents stated that they used delivery logistics services, such as parcel machines, during the COVID-19 pandemic. Most of the respondents (39.5%) stated that the distance they traveled from their place of residence or work to the parcel machine to pick up a parcel was less than 500 m. In turn, every fourth respondent (26.4%) admitted that the distance covered to the parcel machine was from 500 m to 1 km. Similarly, every fifth respondent (16.4%) reported that this distance was over 5 km. For most respondents (71.4%), walking the distance from home or work to collect the parcel took less than 10 min, although there were also those (24.3%) who took 11 to 20 min to reach the parcel machine. Few (0.6%) spent from 41 to 50 min to reach the device location. To collect a parcel, most of the respondents traveled on foot (63.2%) or chose their car as a means of transport (42.9%). Few chose the alternatives of bus (1.2%), bicycle (0.6%) or tram (1.2%).

The respondents, when asked about the reasons for using this form of delivery, i.e., parcel machines in the time of the COVID-19 pandemic, assigned a number of points (from −3 to +3), thus indicating how important a given factor was for them. The collected results show that for the respondents, the most important was time savings (225.71 points), flexibility in choosing the place of delivery (219.71 points), efficiency of delivery and collection (210.29 points), competitive prices for the service (186.86 points) and security of

deliveries (182.29 points). The time of service completion (155.14 points), communication skills (119.71 points) and anonymity of using the delivery service (116.57 points) were less important for the respondents. On the other hand, the company's brand (98.00 points) and the manner and speed of responses to complaints (84.57 points) were of negligible importance. See Figure 3.

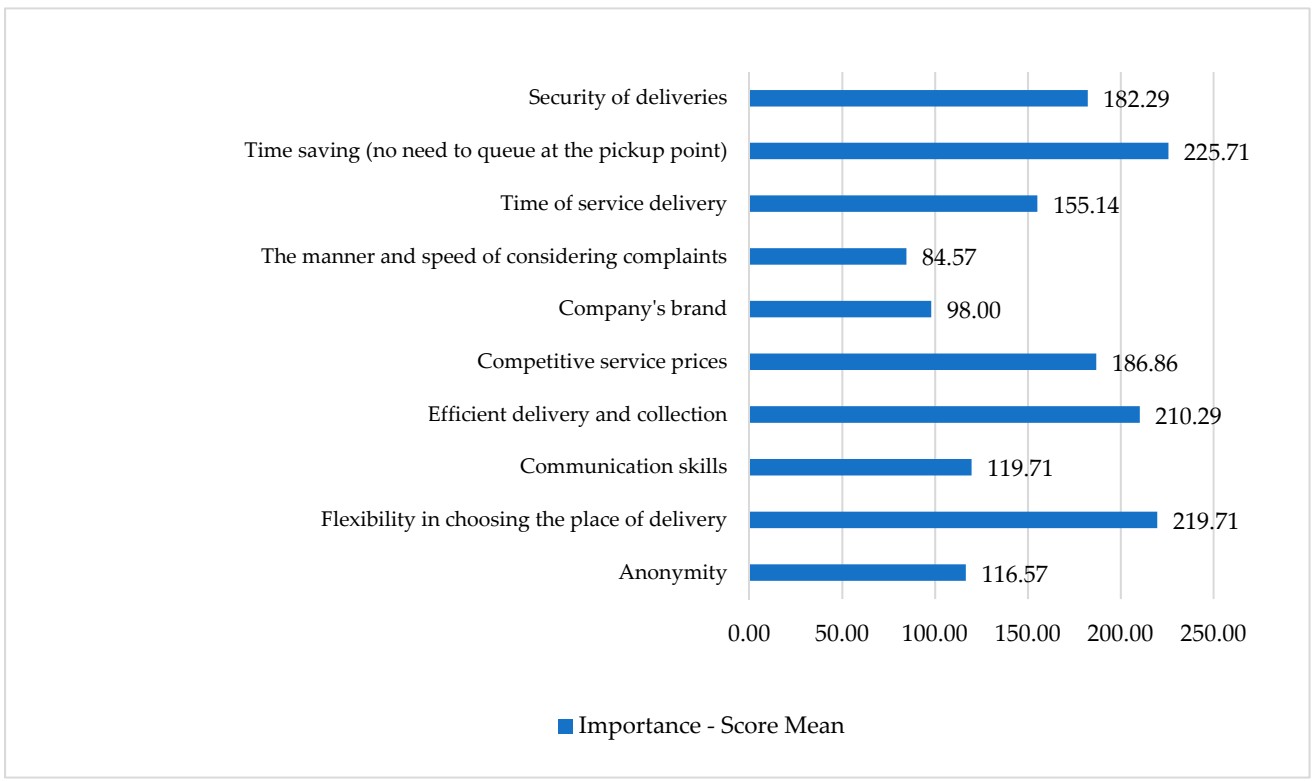

**Figure 3.** The importance of the criteria for selecting parcel machines as a form of delivery logistics service in the last mile area.

For women (114.29 points), saving time was slightly more important than for men (111.43 points), which was illustrated by the awarded scores. Similarly, for women (116.00 points), the flexibility of choosing the place of delivery was slightly more important than for men (103.71 points). Another factor more important for women (96.57 points) than men (85.71 points) was the security of delivery. In turn, for men (102.29 points), the competitiveness of the service price was more important than for women (84.57 points). Similarly more important for men (90.00 points) than for women (65.14 points) was the duration of the service. Another factor more important for men (112.29 points) than for women (98.00 points) was the efficiency of delivery and collection. The existence of the indicated relationships was confirmed by the $\chi^2$ independence test with the strength of the relationship determined by V-Cramer (Table 3).

**Table 3.** Significance of the criteria for selecting parcel machines as a form of delivery logistics service in the last mile area in terms of gender, age and place of residence—the $\chi^2$ independence test with the strength of the relationship defined by V-Cramer.

| Criterion | Gender | | | Age | | | Place of Residence | | |
|---|---|---|---|---|---|---|---|---|---|
| | $\chi^{2\,a}$ | $p^{\,b}$ | $V^{\,c}$ | $\chi^2$ | $p$ | $V$ | $\chi^2$ | $p$ | $V$ |
| Anonymity | 197,990 | 0.001 | 0.549 | 84,890 | 0.001 | 0.161 | 355,703 | 0.001 | 0.329 |
| Flexibility in choosing the place of delivery | 109,523 | 0.001 | 0.408 | 46,537 | 0.004 | 0.133 | 279,249 | 0.001 | 0.326 |
| Communication skills | 235,998 | 0.001 | 0.599 | 148,001 | 0.001 | 0.212 | 340,323 | 0.001 | 0.322 |
| Efficient delivery and collection | 178,123 | 0.001 | 0.520 | 89,659 | 0.001 | 0.213 | 315,607 | 0.001 | 0.400 |
| Competitive service prices | 124,120 | 0.001 | 0.434 | 185,789 | 0.001 | 0.307 | 415,781 | 0.001 | 0.459 |
| Company's brand | 182,240 | 0.001 | 0.526 | 154,368 | 0.001 | 0.242 | 434,636 | 0.001 | 0.406 |
| The manner and speed of considering complaints | 177,698 | 0.001 | 0.520 | 139,687 | 0.001 | 0.188 | 511,093 | 0.001 | 0.394 |
| Time of service delivery | 142,838 | 0.001 | 0.466 | 165,369 | 0.001 | 0.251 | 514,660 | 0.001 | 0.422 |
| Time savings | 15,272 | 0.002 | 0.152 | 84,599 | 0.001 | 0.207 | 244,674 | 0.001 | 0.352 |
| Security of delivery | 99,009 | 0.001 | 0.388 | 177,049 | 0.001 | 0.259 | 399,038 | 0.001 | 0.389 |

[a] $\chi^2$—test value at $\alpha = 0.05$; [b] $p$—asymptotic significance; [c] Relationship strength calculated using V-Cramer.

Innovative solutions such as parcel machines in the area of logistics in the recent COVID-19 pandemic were, according to the respondents, definitely safe for customers and rather in terms of parcel protection (70.2%), in terms of payment (69.9%) and in terms of health protection (63.8%). In turn, for companies, according to the respondents, the use of parcel machines in the time of the COVID-19 pandemic is beneficial for the safety of their employees in terms of health protection (67.2%) and in terms of building trust (65.0%). See Figure 4.

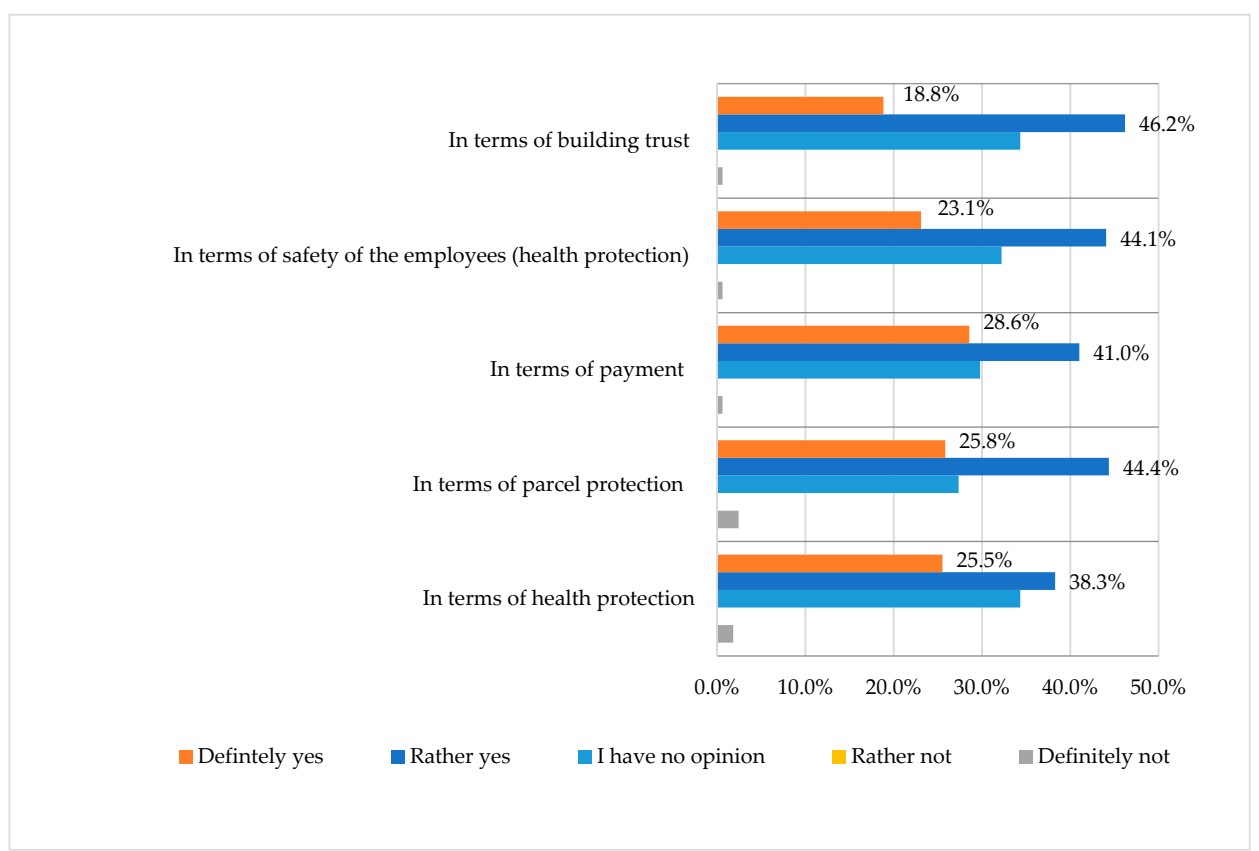

**Figure 4.** Technological innovations in parcel delivery service security with the use of parcel machines during the COVID-19 pandemic.

When asked about shortages of services that the respondents saw among operators of logistics services for parcel delivery, the vast majority (67.9%) said that they did not see any gaps in services. On the other hand, some of the respondents (31.9%) indicated that the services of the delivery operators lacked the possibility of paying in cash, that there was not enough space for packages with large dimensions, or that it was not possible to replace the parcel machine, provide a QR code for another user of the application, extend the holding time for parcels for a fee paid via the Internet, issue SMS messages 3 h before the end of the pickup time, provide a narrator for deaf customers, or provide fresh services.

Moreover, among the problems they (48.9%) encountered at the last mile logistics stage in the area of deliveries to parcel machines, the respondents mentioned queues and insufficient space in parcel machines, redirection to another machine, no possibility of redirection to another parcel locker, no GSM network coverage at a parcel locker, a parcel placed in a locker that was too small, opening another person's locker, inability to open a parcel locker, and the parcel locker software running too slowly.

### 6.4. Future Solutions and Infrastructure Components for Last Mile Logistics

The respondents were asked to indicate those solutions that, in their opinion, will be common among customer logistics services in the near future—10 years. As the most common, the largest group of respondents indicated multiple-use packaging in the logistics circuit (78.4%), voice operation of parcel machines (76.0%), drone deliveries (61.4%) and communication bots in customer service (60.5%). See Figure 5.

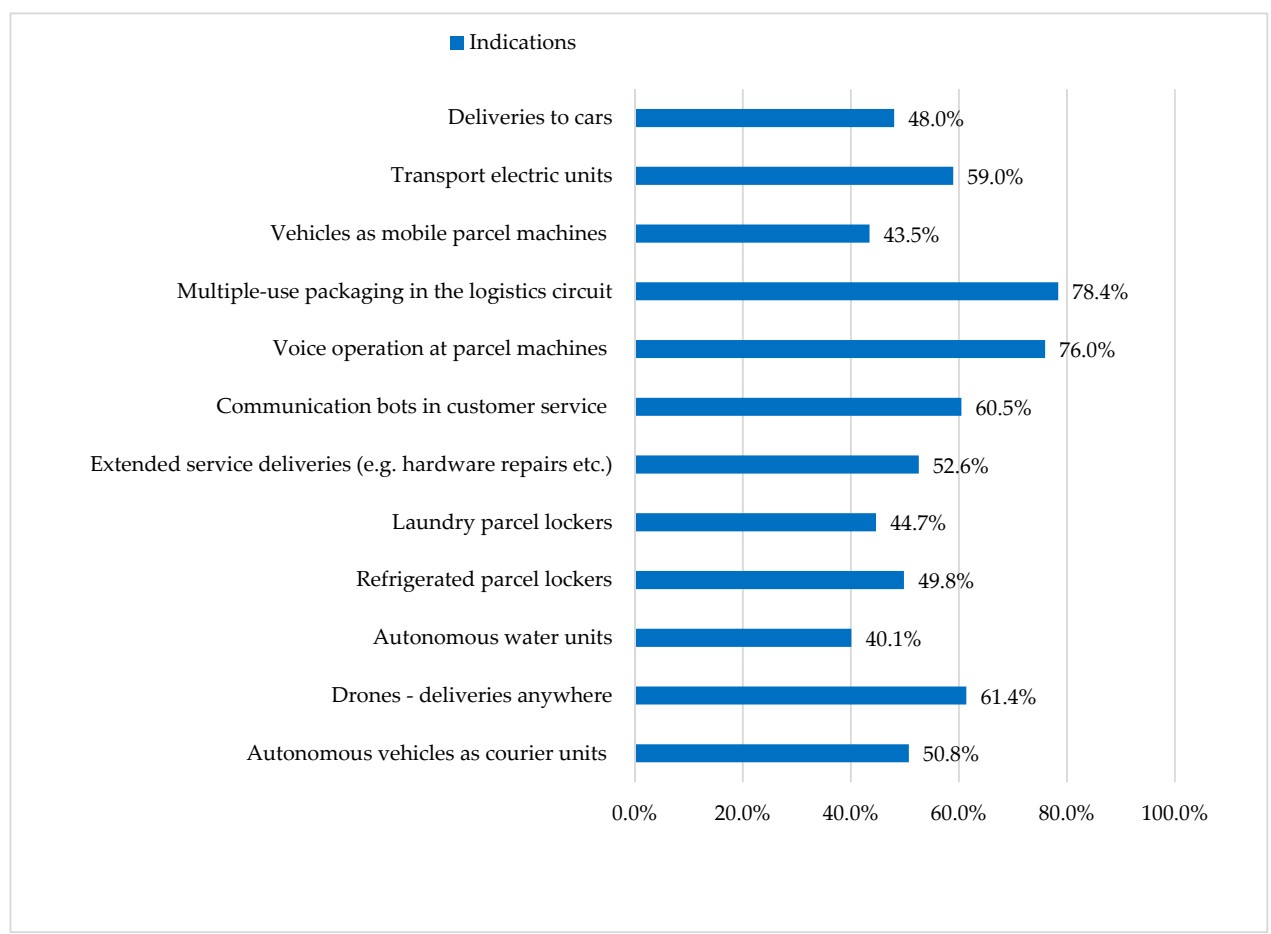

**Figure 5.** The applicability of solutions by companies offering delivery logistics services in the area of customer service in the next 10 years.

Also quite common will be electric transport units (59.0%), extended service deliveries (official documents, equipment repairs) (52.6%), autonomous vehicles as courier units (50.8%), refrigerated parcel lockers (49.8%) and deliveries to cars (48.0%). Research suggests that laundry parcel lockers (44.7%), vehicles as mobile parcel machines (43.5%) and autonomous water units (40.1%) may also be common solutions.

What is more, when asked about the types of innovative technological solutions in the area of last mile service, the respondents clearly indicated reusable packaging in a closed circuit (92.4%), transport electric units (82.7%), voice handling of parcel lockers (81.8%), autonomous vehicles as courier units (76.6%), refrigerated parcel lockers (72.3%), extended service deliveries (69.3%), deliveries to cars (65.0%), communication bots in customer service (65.0%) and laundry parcel lockers (64.4%). In turn, in the last mile service area, in the opinions of the surveyed respondents, there may be autonomous water units (43.2%), vehicles as mobile parcel machines (38.9%) and drones—deliveries to any place (38.6%). See Figure 6.

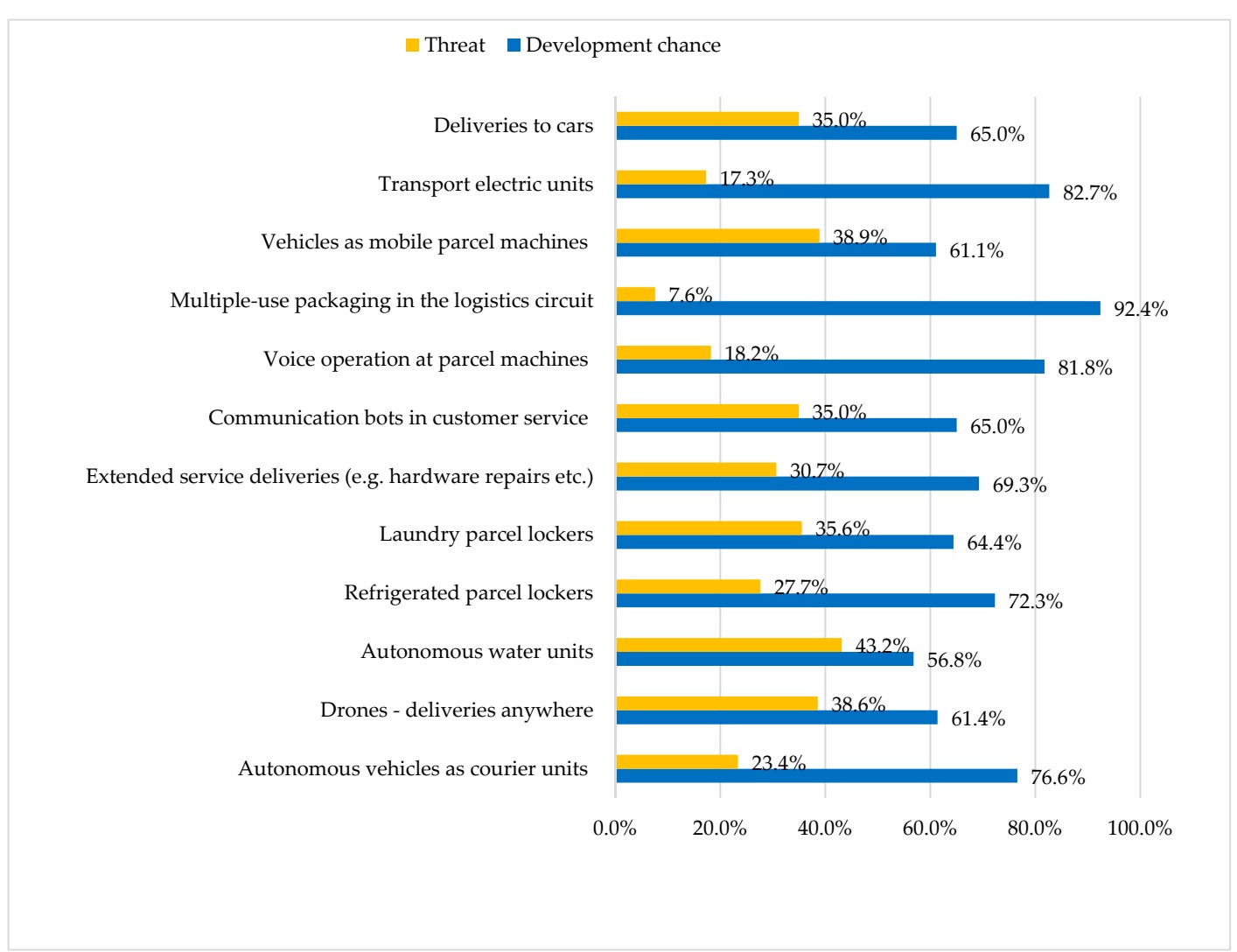

**Figure 6.** Opportunities and threats from introducing innovative technological solutions in the area of last mile service.

According to the respondents, innovative solutions in the field of last mile service may increase the interest in online shopping (94.2%). This direction of development was more often indicated by women (97.7%) than men (90.6%), people younger than 34 years (over 98.0%) and residents of cities with more than 50,000 inhabitants (over 90%). The existence

of the indicated relationships was confirmed by the $\chi^2$ independence test, with the strength of the relationship determined by V-Cramer (Table 4.).

**Table 4.** The impact of innovative forms of delivery logistics services in the last mile area in terms of gender, age and place of residence—$\chi^2$ independence test with the strength of the relationship determined by V-Cramer.

| Variants | Gender | | | Age | | | Place of Residence | | |
|---|---|---|---|---|---|---|---|---|---|
| | $\chi^{2\ a}$ | $p$ [b] | V [c] | $\chi^2$ | $p$ | V | $\chi^2$ | $p$ | V |
| Will increase interest in online shopping | 59,338 | 0.001 | 0.300 | 78,880 | 0.001 | 0.173 | 123,456 | 0.001 | 0.217 |
| Will reduce the need for purchases in traditional stores | 98,653 | 0.001 | 0.387 | 167,399 | 0.001 | 0.252 | 305,264 | 0.001 | 0.341 |
| Both forms of shopping will be equally popular | 95,674 | 0.001 | 0.381 | 100,256 | 0.001 | 0.225 | 178,695 | 0.001 | 0.301 |

[a] $\chi^2$—test value at $\alpha = 0.05$; [b] $p$—asymptotic significance; [c] Relationship strength calculated using V-Cramer.

## 7. Summary and Conclusions

The computer revolution in recent years has transformed the world we live in [66]. The changes have affected the functioning of economies, industries, individual enterprises and citizens. Logistic customer service is a representation of these changes, creating a specific type of bond between enterprises and customers [67]. Its effectiveness and efficiency are based on the use of solutions in the field of new technologies [68], which are more and more often a dominant tool of the competitive struggle in the marketplace. As shown by the results obtained in the course of research on secondary and primary sources, the technology is gaining more and more recognition among customers—users.

Technologies enabling the implementation of Industry 4.0 include, among others: advanced production solutions, additive manufacturing, augmented reality, simulation, horizontal and vertical integration, deep learning, artificial intelligence, cloud computing, Internet of Things, cyber security, big data and analytics [69,70]. Modern automation and robotization technologies implemented in the information and organizational structures of enterprises [71] contributed to and still have an impact on the improvement of processes related to design, production, logistics, supply chains and customer service. When the supply chain covers the entire product life cycle (warranty, regeneration, recycling, disposal, etc.), the impact of logistic flows on the network structure is significant [72]. In this context, the concept of Industry 4.0 [73] emphasizes systemic functioning in the network, the essence of which is the integration of people and digitally controlled machines, with the simultaneous use of the Internet and information technologies. The benefits of implementing, and operating in accordance with, this concept result primarily from coordination activities. The demand for effective coordination of logistics processes, including customer service, going beyond a given company is treated as a background of high technological pressure in the practices of enterprises [74].

The CEP industry has played a significant role in e-commerce activity during the COVID-19 pandemic, determining the assessment and finalization of transactions. The experiences of the respondents—customers of CEP services offered by enterprises operating in the Polish market—indicate the knowledge of the companies themselves and the spectrum of delivery logistics formats they offer in the last mile area.

Customers using innovative solutions in the area of CEP services during the COVID-19 pandemic were satisfied with the last mile service formats. The reasons for using the services of advanced automation solutions (PUDO) were mainly determined by time savings, flexibility in choosing the place of delivery, the effectiveness of delivery and collection, competitive services and safety. These components represent compliance prerogatives for the adoption of technology solutions by CEP customers in the last mile service area.

Innovation in the technologization of delivery services in the context of last mile logistic services will determine the future of the CEP industry. It is important not only for the speed of delivery, but also for the flexibility and freedom of decision-making, as well as health and safety. As part of this trend, the respondents (customers) indicated such

technological solutions as reusable packaging, voice control of parcel machines, deliveries by drones, communication bots in customer service, extended service deliveries (official documents, equipment repairs), autonomous vehicles as courier units, refrigerated parcel lockers, and deliveries to cars. The experiences gained by customers during the COVID-19 pandemic are already resulting in an increase in customer expectations regarding the pro-innovation of delivery services offered by the CEP industry in the area of last mile logistics.

In the course of this research and analysis of its results, the research hypotheses were positively verified, which allowed us to confirm the research thesis. Thus, the original goal of the article indicated in the introduction, assuming the impact of the COVID-19 pandemic on the quality of services offered by enterprises in the CEP industry in the context of innovative technological solutions at the last mile stage in Poland, has been achieved. This study is the first in a planned series of studies on technological innovation in last mile logistics process services in electronic commerce, and the authors plan to extend their research to other European countries.

To sum up, in the course of deliberations, it was confirmed that in the near future, the provision of logistic services by the CEP industry should be associated with the necessity to increase the level of experience in service and the development process based on technological innovations. This is what customers expect. It is also indicated by changes observed in the market. After the end of this research, it was observed that the stock prices of companies that benefited from the pandemic had returned to their pre-pandemic levels. However, in the Polish market, where the research was conducted, the situation has remained stable, as enterprises from the CEP industry continue to record high volumes of demand for their services generated by online shopping. It is likely that customer experiences will dictate further changes in the industry that will be possible to note in subsequent studies.

**Author Contributions:** All authors contributed to conceptualization. Methodology, Ł.S. and K.K.-M.; software, P.M.; validation, M.B. and P.M.; formal analysis, M.B. and T.S.; resources, K.K.-M. and Ł.S.; data curation, P.M.; writing—original draft preparation, Ł.S., K.K.-M. and M.B.; writing—review and editing, Ł.S., K.K.-M., M.B., P.M. and T.S.; visualization, P.M.; supervision, M.B.; project administration, K.K.-M. and P.M.; funding acquisition, Ł.S., K.K.-M., M.B. and P.M. All authors have read and agreed to the published version of the manuscript.

**Funding:** This research was funded Jagiellonian University, Faculty of Management and Social Communication, Jan Kochanowski University of Kielce, Cracow University of Economics (Doskonałość Badawcza nr 71/ZZN/2021/DOS), Lodz University of Technology.

**Institutional Review Board Statement:** Ethical review and approval were waived for this study, due to fact that the entire research was anonymous.

**Data Availability Statement:** Not applicable.

**Conflicts of Interest:** The authors declare no conflict of interest.

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
