# Peer review of "Last Mile Logistics Innovations in the Courier-Express-Parcel Sector Due to the COVID-19 Pandemic"

_sustainability, doi:10.3390/su14138207_

Round 1

Reviewer 1 Report

The purpose of the article is very current and the results obtained clearly demonstrate the expectations of customers for the CEP industry, justifying the importance of investiments in technological solutions.

I would like to make just a few suggestions:

  1. in my analysis, the content of the Introduction could be presented a little more succinctly, leaving some principles to be discussed after the research results and before the conclusion.
  2. As I am a doctor, I don't feel qualified to judge about the English language and style used in a non-exclusive scientific article directly related to the health area.
  3. There are some typos that need to be corrected, such as, always use SARS-CoV-2. Sometimes it was typed: SAR-CoV-2, SARS Cov-2, SARS-COV 2.
  4. I also consider it necessary to describe each of the abbreviations whenever they appear for the first time in the text, for example: CEP (line 18), ICT (line 49), PLN (line 198).
  5. I didn't understand the sentence in lines 88 and 89 of the text.

Author Response

Dear Reviewer,

Thank you for your time spent to read our article and evaluate our work. We found your comments extremely valuable. Your detailed recommendations have been considered and as result we upload updated version of our article for another review. As your first review was quite possitive with some minor changes (which have been implemented) we are convinced now our article is ready to be published. Again thank you for your time and effort to make our article improved.

Best Regards,
Authors

Reviewer 2 Report

The manuscript seems aimed to discuss last mile logistics and impact of the pandemic.

The results read fit for the title and analysis aim, however, the introduction of trade market, crisis and prediction of recession made me lost for a while, thinking of stock market…., a quite different topic. While the discussion section did not talk much on the study result and stock market or economy. Please explain.

The results summarized e-commerce during pandemic. While from the recent stock market in the US, many of the companies benefited from pandemic (online shopping, virtual meeting websites/software) have jumped back to their before-pandemic price. May be an indication of customers’ consumption needs  shifting from virtual (partially) back to in-store (experience). How does the analysis from this manuscript associate with this coming change?

Minor:

Line 17: typo, “SAR-CoV-2”should be “SARS-CoV-2”

Missing literature citation: PMCID: PMC8435763

Author Response

(The authors gave the same response as above.)

Reviewer 3 Report

Dear Authors,
The study aims at investigating the " Last mile logistics innovations in the Courier-Express-Parcel sector due to the COVID-19 pandemic". 
I am happy to inform you that I have accepted your manuscript and will recommend it for publication without further changes. Congratulations. I look forward to reading it online.
Thank you for the opportunity to let me contribute a small part to your publication. 

Author Response

Dear Reviewer,

Thank you for your time spent to read our article and evaluate our work. On behalf of author’s team I express our BIG THANK YOU for such possitive review. We also look forward to see article published (online) as soon as possible.

Best Regards,
Authors